# Attenuation of a Novel Goose Parvovirus Strain NMG21 via Serial Cell Passage

**DOI:** 10.3390/v17050618

**Published:** 2025-04-25

**Authors:** Jing Yang, Dalin He, Bingrong Wu, Yun Yan, Yupei Zhang, Jiaping Zhou, Feng Wei, Youjiang Diao

**Affiliations:** 1College of Agricultural Science and Technology, Shandong Agriculture and Engineering University, Ji’nan 250100, China; yj9210@163.com (J.Y.); yanyun__yy@163.com (Y.Y.); 18615643559@163.com (Y.Z.); jiaping1022@163.com (J.Z.); 2College of Veterinary Medicine, Shandong Agricultural University, Tai’an 271018, China; dlhe1230@163.com (D.H.); brongwu2021@163.com (B.W.); 18854888209@163.com (F.W.)

**Keywords:** N-GPV, BADS, DEFs, pathogenicity, virulence

## Abstract

The novel goose parvovirus (N-GPV), responsible for beak atrophy and dwarfism syndrome (BADS), has caused significant economic losses in China’s duck-raising industry. In this study, a highly virulent N-GPV strain NMG21 was serially passaged in duck embryo fibroblast cells (DEFs). The virus titers and virulence of selected passages were evaluated in 1-day-old ducklings. An increased virus titer was observed at the 5th passage (P5). Compared with the parent strain NMG21, the P35 (NMG21-35 strain) has a clear decrease in pathogenicity for ducklings, with less tissue damage. The NMG21-35 also exhibited relatively lower tissue replication rates and higher antibody levels. Collectively, the virulence of N-GPV strain NMG21 was reduced via serial passage in DEFs for 35 passages. Our research successfully prepared a N-GPV attenuated variant which might serve as a potential live vaccine candidate against N-GPV infection. Developing a live attenuated vaccine candidate against N-GPV infection in China is crucial for mitigating the economic impact of N-GPV on the duck industry.

## 1. Introduction

The fight against zoonotic viral infections of wild and domestic animals occupies an important place in human activities [1,2]. In 2015, a novel duck infectious disease characterized by short beak with protruding tongue, podgy legs, growth retardation, and watery diarrhea emerged in multiple duck facilities throughout China [3]. Based on these alterations, the disease was diagnosed as novel goose parvovirus (N-GPV)-triggered beak atrophy and dwarfism syndrome (BADS) [4]. Moreover, the disease could increase the feed conversion ratio of the infected flocks, resulting in great economic losses to the duck-raising industry [3]. N-GPV, similar to genus *Anseriform dependoparvovirus 1*, is a 5-kb-genome DNA virus having a single strand [5]. The genome contains two major open reading frames (ORFs): the left ORF that encodes for the regulatory (Rep) protein, and the right ORF that encodes for three capsid proteins, VP1, VP2, and VP3 [6].

Initially reported in France and Poland during the 1970s, the disease re-emerged in China in 2015, manifesting in clinical signs such as shortened beaks, protruding tongues, fragile tibiae and pteroids, as well as growth retardation in ducks [7]. N-GPV remains prevalent in China and is often found in mixed infections with other viruses [8,9]. Currently, GPV vaccines and anti-GPV egg yolk antibodies are used to control BADS [10], which may be the main reason for the lack of effective vaccines to prevent and control BADS. However, differences in key amino acid sequences of N-GPV may reduce the efficacy of these biological products in protecting ducks against BADS [11,12].

The lack of an effective vaccine is a major factor contributing to the high prevalence of N-GPV. To develop a live attenuated vaccine candidate for N-GPV, the NMG21 strain was attenuated through serial passage in duck embryo fibroblast cells (DEFs) for 35 passages. The attenuated strains were tested in 1-day-old ducklings, and the severity of clinical signs, histopathological lesions, viral shedding, antibody levels, and replication kinetics were evaluated. The results demonstrated that serial passage in DEFs effectively attenuated the virulence of the NMG21 strain.

## 2. Materials and Methods

### 2.1. Virus, Animals and Ethics Statement

N-GPV strain NMG21 was isolated and preserved in our laboratory. The 30 9-day-old duck embryos and 180 1-day-old ducklings used in this study were purchased from a hatchery in Tai’an, Shandong. All animals and embryonated eggs in this study were initially free of the specific pathogen we studied. The ducklings were housed in SPF animal isolators with negative pressure ventilation and provided with ad libitum access to food and water [7]. The experimental protocol was approved by the Ethical Committee for animals in research of the Shandong Agricultural University, China, and adhered to the ARRIVE guidelines.

### 2.2. Adaptation and Serial Passages in DEFs

DEFs were isolated from 9-day-old duck embryos and cultured in Dulbecco’s modified Eagle’s medium (DMEM/F-121:1) (01-172-1ACS, BI, Kibbutz, Beit Haemek, Israel) supplemented with 10% fetal bovine serum (FBS) (04-001-1ACS, BI, Kibbutz, Beit Haemek, Israel), 100 μ/mL penicillin, and 100 μg/mL streptomycin (P1400, Solarbio Science & Technology Co., Ltd., Beijing, China) at 37 °C with 5% CO_2_. The NMG21 strain was serially passaged in DEFs for a total of 35 passages, with passages occurring every 4 to 6 days.

### 2.3. Virus Titration of Different Passages

The NMG21 strain and its passages (NMG21-5, NMG21-15, NMG21-25, and NMG21-35) were titrated in 9-day-old duck embryos. Viruses in each passage were subjected to a series of 10-fold dilutions from 10^−1^ to 10^−7^, and egg infectious dose 50 (EID_50_) was determined following the method of Reed and Muench (1938). PCR was used to detect viral infection of embryonated eggs, and lesions such as hemorrhages, congestion or stunting, and death were recorded. Allantoic fluids were collected and detected for the presence of the N-GPV virus with PCR in the lab.

### 2.4. Animal Experiment Design

To evaluate the pathogenicity of passaged N-GPV strains in ducklings, clinical signs and body weights were monitored for 28 days. A cohort of 180 one-day-old ducklings was randomly stratified into six experimental cohorts (*n* = 30/group), designated as NMG21, NMG21-5, NMG21-15, NMG21-25, NMG21-35, and a negative control group. Experimental cohorts received intramuscular administration of 0.2 mL N-GPV culture supernatant, whereas control subjects underwent sham inoculation with an equivalent volume of sterile phosphate-buffered saline (PBS). All groups were maintained in controlled isolation units under SPF conditions with standardized nutritional protocols. Post-inoculation observational parameters included clinical morbidity indices, macroscopic pathological manifestations, and histopathological alterations in tissue architecture.

### 2.5. Sample Collection

On days 1, 7, 14, 21, and 28 post-inoculation (dpi), six ducklings from each group were randomly selected for weighing, euthanized with intravenous pentobarbital sodium (New Asia Pharmaceutical, Shanghai, China), and necropsied. Tissues including the heart, liver, spleen, lung, kidney, bursa, thymus, pancreas, brain, proventriculus and duodenal, femur, and tibia (volume of tissue specimens is 1 cm × 1 cm × 1 cm) were collected for histopathological examinations and viral load analyses. Blood samples and cloacal swabs were also collected for viral DNA detection.

### 2.6. Histopathology

Tissues were fixed in 10% formaldehyde solution, embedded in paraffin, and sectioned at 4 μm. Sections were stained with hematoxylin and eosin (HE) for histological analysis [13].

### 2.7. DNA Extraction and Quantitative Reverse Transcription PCR (qRT-PCR) Analysis

Viral DNA was isolated from tissue specimens through standardized phenol-chloroform extraction protocols employing DNAiso Reagent (Cat. #9770Q, Takara Bio, Kusatsu, Japan) in accordance with manufacturer protocols. Nucleic acid quantification was conducted via ultraviolet spectrophotometry using a DeNovix DS-11 microvolume spectrophotometer (DeNovix Inc., Wilmington, DE, USA), with absorbance ratios (260/280 nm) verifying macromolecular purity. The qRT-PCR amplifications were performed in 20 μL reaction volumes containing the TaKaRa One-Step PrimeScript RT-PCR Master Mix (Takara Bio, Dalian, China) on a Roche LightCycler 96 thermal cycler (Roche Diagnostics, Basel, Switzerland). Amplification parameters were followed a proprietary TaqMan-based qRT-PCR assay optimized in our laboratory.

### 2.8. Serology

After centrifugation (3000× *g* for 10 min), serum samples were separated for the test of N-GPV specific antibodies, to assess the humoral immune response of hosts to N-GPV infection. The indirect competitive ELISA was performed in accordance with a previously described method [14].

### 2.9. Statistical Analysis

Data were analyzed by GraphPad Prism version 5.0 software (GraphPad Software Inc., San Diego, CA, USA) and shown as mean values ± SEM. To investigate the differences among assays, the data were analyzed using ANOVA via the method of Bonferroni (SPSS Statistics 17.0, IBM, Armonk, NY, USA). *p* < 0.05 was considered to be statistically significant.

## 3. Results

### 3.1. Attenuated by Serial Passage

The embryos inoculated with NMG21-5 died by the third day, showing severe hemorrhaging and beak atrophy. By the 15th passage, embryo mortality decreased, and embryos exhibited free of visible lesions and beak atrophy. By the 35th passage, embryos exhibited free of visible lesions and normal beak development (Figure 1). The EID_50_ values for NMG21, NMG21-5, NMG21-15, NMG21-25, and NMG21-35 were 10^−7^/0.2 mL, 10^−5^/0.2 mL, 10^−4.7^/0.2 mL, 10^−4.2^/0.2 mL, and 10^−3^/0.2 mL, respectively (Table 1).

### 3.2. Clinical Signs

Ducklings in the NMG21, NMG21-5, NMG21-15, and NMG21-25 showed clinical signs and began dying between 2 and 7 dpi, while NMG21-35 group showed clinical signs at 10 dpi. Severe clinical signs, including paralysis (Figure 2a), short beak (Figure 2b), tongue swelling (Figure 2c), and growth retardation (Figure 2e), were observed in the NMG21 group. Feather loss was only observed in the NMG21-35 group (Figure 2d). The pathogenicity of the NMG21 strain to the ducklings gradually decreased with the cell passages (Figure 2f–h). The death date of ducklings in groups NMG21, NMG21-5, and NMG21-15 were concentrated at 4–7 dpi, and the mortality rate ranged from 26.7–40%. Ducklings in groups NMG21-25 and NMG21-35 died at 11 dpi, and the mortality rate was 6.7% and 2.8%, respectively (Figure 2i).

### 3.3. Gross Lesions

Postmortem examination revealed that the main gross lesions of the infected ducklings included hemorrhage of the liver, hemorrhage of the lung, and hemorrhage and enlargement of the spleen (black arrows) (Figure 3). With the cell passages, the pathogenicity of passaged N-GPV strain to ducklings became weaker. No gross lesions were observed in any ducklings in the control group. By measuring the beak length and tongue length of ducklings at different days of age in the control group, the standard curve was established between the beak length (cm) and the tongue length (cm) (R^2^ = 0.992), with a regression line revealing an average intercept and slope of 0.1173 and 0.989, respectively. The incidence of beak atrophy in different infected groups was measured at random. The incidence of NMG21 and NMG21-5 was 95.83% (Figure 4a,b), that of NMG21-15 and NMG21-25 was 91.67% (Figure 4c,d), and that of NMG21-35 was 87.5% (Figure 4e). And beyond that, retardation of bone growth was observed in the inoculated groups, including shorter length and lighter weight of the femur (Figure 5a–c) and tibia (Figure 5d–f). The influence on the bone development of ducklings decreased with the cell passages.

### 3.4. Histopathology

In the NMG21-infected group, significant histological lesions (green arrows) were observed in the liver, spleen, lung, and duodenal (Figure 6). The histopathologic changes in spleen, and the duodenal manifested significantly. Livers were displayed histopathologic changes, such as varying degrees of focal necrosis of hepatocytes, hemorrhage of intranuclear inclusion bodies, hepatic cell swelling and steatosis, and various sizes of fat droplets in the cellular protoplasm. Additionally, there were significant pathological changes in the spleen and lungs of ducklings, including a large number of apoptotic and necrotic areas visible in the spleen, and mild hemorrhage in the lung along with a large number of neutrophil nuclear fragmentations, light coloration, unclear boundaries of the lung lobules, and broken capillary arms resembling emphysema. Meanwhile, HE staining showed lesions characterized by significant atrophic exfoliation of intestinal villi. Histopathology indicated lower pathogenicity of the attenuated strains, which showed mild emphysema of lung cells, mild swelling of liver cells, hyperplasia of spleen cells, and atrophic exfoliation of intestinal villi accompanied by inflammatory cell exudation.

### 3.5. Viral Loads in Tissues

The qPCR analysis of N-GPV genome copy numbers in infected ducklings revealed active viral replication across all 10 tissues throughout the experimental period (7, 14, 21 dpi), as shown in Figure 7a–c. The viral load was highest in the heart, liver, proventriculus, and intestine, while tissues such as the spleen, kidney, pancreas, and others showed relatively lower viral loads. In all five infection groups, the viral load pattern remained consistent across the tissues. However, the NMG21-35 group exhibited the lowest viral copy numbers compared to the other groups. No viral DNA was detected in the control ducklings during this study.

### 3.6. Viral Shedding in the Blood and Cloaca

By detecting viral DNA in the blood and cloaca at 1, 7, 14, 21, and 28 dpi, we established the viral shedding pattern after N-GPV strains were passaged and infected. The results of N-GPV copy numbers in blood are shown in Figure 7d. The virus copy number in the blood of NMG21-15 and NMG21-25 groups could be detected as early as 1 dpi and reached the replication peak at 14 dpi, while the other three groups reached the replication peak at 7 dpi. Afterward, the viral load decreased over time. The results of N-GPV copy numbers in cloacal swabs are shown in Figure 7e. The changes in viral copy numbers in cloacal swabs were similar across all five infection groups, with all groups peaking at 7 dpi.

### 3.7. Antibody Response

N-GPV-specific antibody levels in the serum were detected throughout the experiment. As Figure 7f shows, except for the NMG21-25 group, which reached the highest level at 14 dpi, the NMG21, NMG21-5, and NMG21-15 groups reached double peaks at 7 dpi and 21 dpi. Subsequently, antibody level declined with time. The induced antibody response in the NMG21-35 group rose rapidly and reached the peak at 14 dpi. Although it decreased thereafter, the differences still remained compared with that of the other four groups.

## 4. Discussion

Waterfowl parvovirus induces significant morbidity and mortality in geese and Muscovy ducks, with mortality rates ranging from 10% to 80% [15,16]. The disease is associated with a morbidity rate of 10% to 50%, resulting in considerable economic losses within China’s duck industry [16]. Currently, vaccination remains the most effective measure against N-GPV. Although a few studies have demonstrated N-GPV was pathogenic to Cherry Valley ducklings, there are no effective treatments or any vaccines against N-GPV infection.

In this study, N-GPV NMG21 was passaged regularly to a total of 35 passages in DEFs, and the EID_50_ of P5, P15, P25, and P35 was determined in vitro. The titers of P35 were significantly higher than those of other passages, indicating that P35 had better adaptation in cells in vitro. Usually, adaptation in cells coincides with attenuation of the viruses, such as Tembusu virus (TMUV) strain P3, in 2-day-old Pekin ducks [17,18,19,20]. The higher fitness of P35 in DEFs might suggest the changed pathogenicity in N-GPV. In the present study, attenuated N-GPV variants were intramuscularly inoculated to ducklings to compare the tissue tropism and pathogenicity to its parent strain.

After infection, compared with the NMG21 strain, P5, P15, and P25, P35 caused slight clinical signs in ducklings, suggesting that the pathogenicity of N-GPV changed through cell passage. However, gross lesions caused by N-GPV-attenuated strains, although much milder and in fewer organs than those in ducklings infected with the NMG21 strain, were still detectible at necropsy. Interestingly and importantly, the clinical findings for feather loss of P35-inoculated ducklings were found. Previous studies have provided molecular evidence for GPV-related abnormal molting in Pekin ducks [8]. Additionally, Yang et al. revealed the high coinfection rate of the N-GPV and DuCV in collected 540 duck feather sac samples of sick ducks with typical feather shedding syndrome from 27 Cherry Valley duck flocks in East China [21]. PCR for the detection of DuCV were performed for this purpose. Negative DuCV suggests that the appearance of the feather shedding in P35 inoculated ducklings might be closely related to the virulence of N-GPV.

The NMG21-35 attenuated strain demonstrated the ability to elicit a stronger humoral antibody response upon inoculation. This finding indicates that this strain may be a valuable candidate for inclusion in vaccination programs under field conditions, potentially mitigating the economic losses associated with N-GPV infections in commercial duck-raising industry. Nevertheless, as a candidate vaccine, in addition to its pathogenicity, the protection efficiency of NMG21-35 strain against epidemic strains should also be evaluated. The emphasis of future research will be on the immunity assessment of the NMG21-35 strain to challenge with epidemic N-GPV.

The findings presented here collectively confirm that the P35 strain generated in this study exhibits low pathogenicity in ducklings and may serve as a promising candidate for a novel N-GPV vaccine. However, several critical questions remain to be addressed. For example, the attenuation of a NMG21-35 strain might be the result of a combination of multiple mutations along the genome, and can occur via multiple molecular mechanisms, so further genetic investigation will be necessary to determine the key mutations responsible for the attenuation in the NMG21-35 strain. Moreover, whether individual or a combination of genetic changes of N-GPV strains alter viral infectivity, pathogenicity, and replication efficiency need further examination by reverse genetics technology. Addressing these questions is crucial for the development of a safe and effective live-attenuated vaccine for N-GPV control. In conclusion, our study successfully generated an attenuated N-GPV variant, P35, which demonstrated the highest virus titers following serial passage of the parental NMG21 strain. Notably, inoculation of ducklings with P35 via intramuscular injection resulted in minimal clinical symptoms and pathological changes.

## Figures and Tables

**Figure 1 viruses-17-00618-f001:**
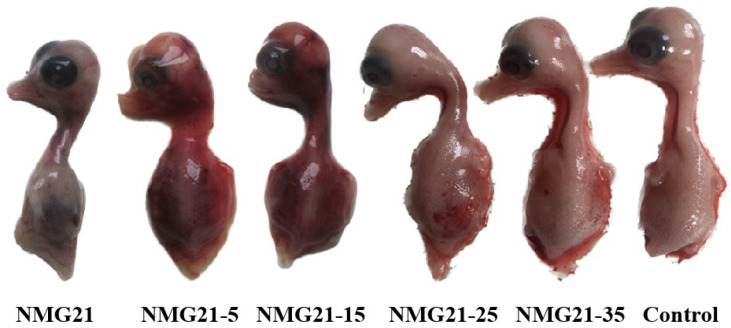
Typical duck embryo lesion caused by NMG21, NMG21-5, NMG21-15, NMG21-25, and NMG21-35 strains.

**Figure 2 viruses-17-00618-f002:**
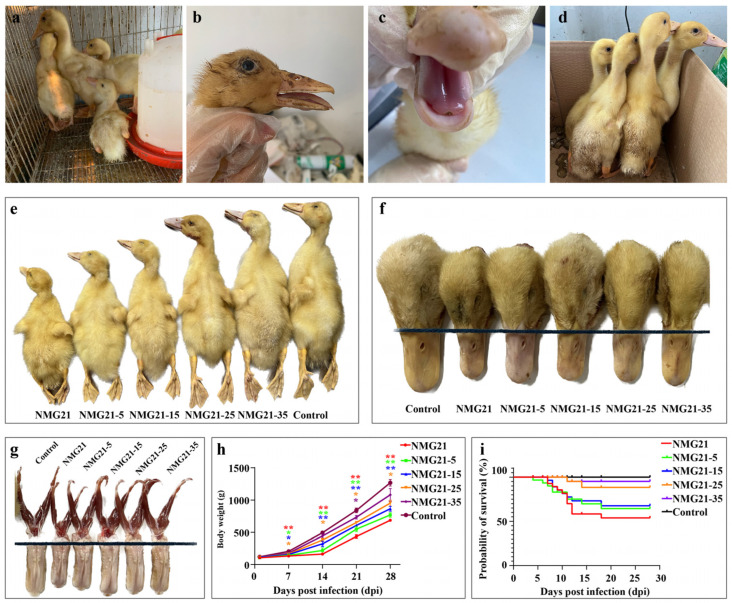
(**a**) Representative clinical symptoms showing paralysis. (**b**) Representative clinical symptoms showing short beak. (**c**) Representative clinical symptoms showing tongue swelling. (**d**) Representative clinical symptoms showing feather shedding of ducklings inoculated with NMG21-35 strain. (**e**) Individual size difference of ducklings in each group. (**f**) Differences in beak length between groups. (**g**) Tongue wear in each group. (**h**) Weight gain loss. (**i**) Survival percentage. Data are expressed as mean values ± SEM (*n* = 3): * *p* < 0.05, ** *p* < 0.01 vs. control.

**Figure 3 viruses-17-00618-f003:**
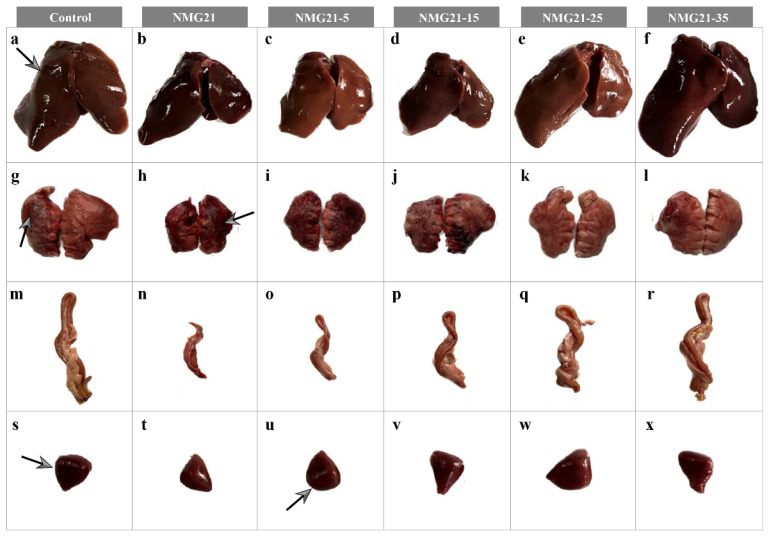
Representative gross lesions caused by N-GPV. (**a**–**f**) Liver. (**g**–**l**) Lung. (**m**–**r**) Duodenum. (**s**–**x**) Spleen.

**Figure 4 viruses-17-00618-f004:**
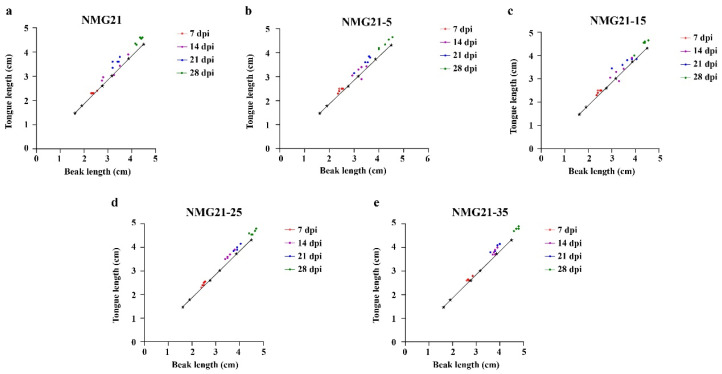
The incidence of beak atrophy in different infected groups. (**a**) NMG21-infected group. (**b**) NMG21-5-infected group. (**c**) NMG21-15-infected group. (**d**) NMG21-25-infected group. (**e**) NMG21-35-infected group.

**Figure 5 viruses-17-00618-f005:**
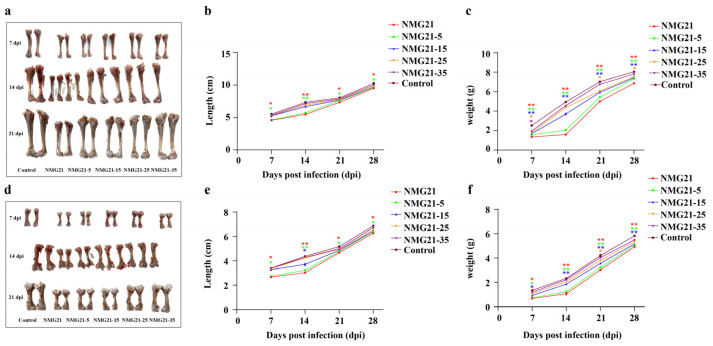
(**a**) Representative pictures of tibia from different infected groups. (**b**) Trends in tibia length between different infected groups. (**c**) Trends in tibia weight between different infected groups. (**d**) Representative pictures of femur from different infected groups. (**e**) Trends in femur length between different infected groups. (**f**) Trends in femur weight between different infected groups. Data are expressed as mean values ± SEM (*n* = 3): * *p* < 0.05, ** *p* < 0.01 vs. control.

**Figure 6 viruses-17-00618-f006:**
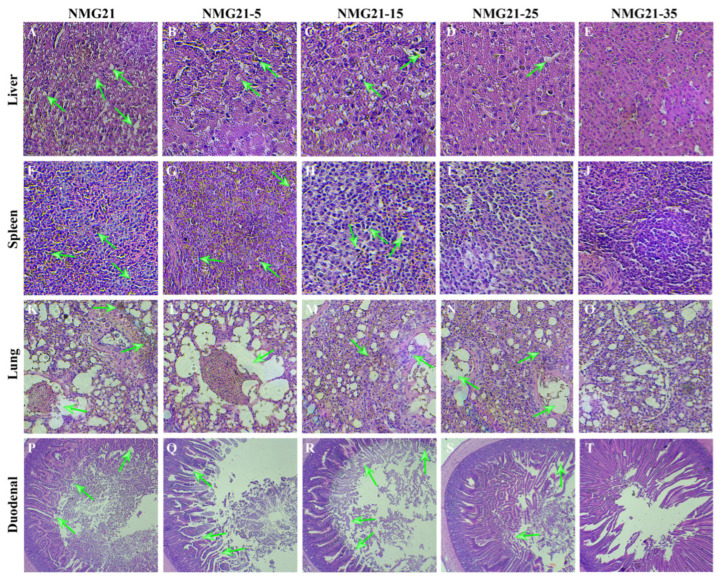
Histopathology lesions of organs induced by N-GPV infection. Tissues were stained with hematoxylin and eosin (HE). (**A**–**E**) Liver (HE, 200×). (**F**–**J**) Spleen (HE, 200×). (**K**–**O**) Lung (HE, 200×). (**P**–**T**) Duodenum (HE, 50×).

**Figure 7 viruses-17-00618-f007:**
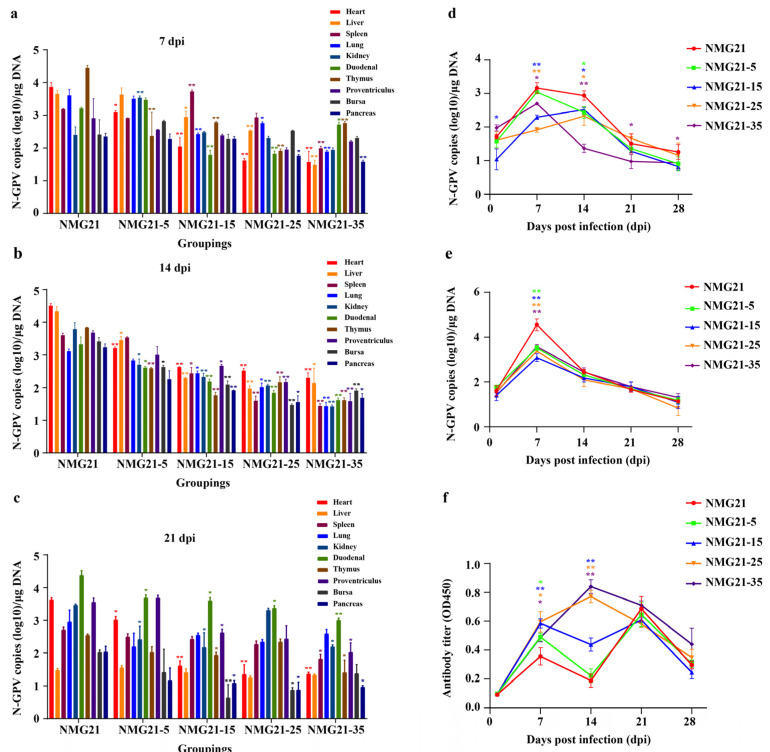
(**a**) Dynamics of N-GPV loads in tissues at 7 days post-inoculation. (**b**) Dynamics of N-GPV loads in tissues at 14 days post-inoculation. (**c**) Dynamics of N-GPV loads in tissues at 21 days post-inoculation. (**d**) Patterns of viral shedding in blood. (**e**) Patterns of viral shedding in cloacal swabs. (**f**) Dynamics of antibodies against N-GPV in the serum of ducklings after artificial infection. Data are expressed as mean values ± SEM (*n* = 3): * *p* < 0.05, ** *p* < 0.01 vs. NMG21.

**Table 1 viruses-17-00618-t001:** The virus titers and virulence of different serial passages of NMG21 strain.

Generation	P0 (NMG21)	P5 (NMG21)	P15 (NMG21)	P25 (NMG21)	P35 (NMG21)
EID_50_ (0.2 mL)	10^−7^	10^−5^	10^−4.7^	10^−4.2^	10^−3.7^

## Data Availability

The data that support the findings of this study are contained within the article. The data and materials that support the findings of this study are available from the corresponding author upon reasonable request.

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
