# Peer review of "Attenuation of a Novel Goose Parvovirus Strain NMG21 via Serial Cell Passage"

_viruses, 2025, doi:10.3390/v17050618_

Round 1
Reviewer 1 Report
Comments and Suggestions for Authors
Authors generated and tested attenuated nGPV strain of ducklings.
The major minus is that authors did not perform sequencing on viruses especially parental strain and p35 mildly attenuated strain. From results it seems that still the p35 induces pathogenicity in ducklings. We do not know how many mutations and where in the genome were introduced, also in discussion it should be discus possible reversion to WT. It is standard procedure to sequence genome of potential vaccine.
No proper statistical analysis was performed, you cannot see it on graphs and figures.
Therefore I recommend major revision.
General comment:
N-GPV i think NGPV or nGPV is more commonly used so no dash.
Specific comment:
Line 31 up-regulate cange to increased (up regulat usually implies genes)
line 34 omit negative sense it is relevant for RNA viruses only
Figure 2 description in legend mixed up
Figure 3 what is an arrow?
Line 147-148 From the pictures I can not see any haemorhages.
Fig 3 Apart from liver pictures, everything looks the same what is the point of this figure?
Figure 5. Are this differences statistically important?
5f wrong description in graph.
Line 177what does it mean “red staining”
Figure 6 what are the arrows? Also see general comment, arrowheads should be black, because it fades with background picture
Fig 7 again statistics! Add days p.i to graphs for quicker understanding for the reader
d- viremia (no blood shering)
238 this sentence is not clear :as a newly n_GPV?)
In discussion you mention feather abnormalities, it should be mention in results section.
Disscusion part does not discus the results thorouly, what is the basic of attenuation (why there is antibody drop in day 14 in wt and other mutants).
Also some figures are unreadable , like anatomopathological lesions, histopathology, they are too small. Maybe do not show pictures from every challenge group, just write about it in manuscript and show magnifications.
Author Response
Dear Editor and Reviewers:
Many thanks for your professional evaluation of our manuscript entitled “Attenuation of an original novel goose parvovirus strain NMG21 via serial cell passage” (ID: ID: viruses-3573035). On behalf of my co-authors, we would like to express our great appreciation to the editor and reviewers. We studied the comments carefully and have made corrections that we hope meet with approval. The main corrections in the paper and the responses to the reviewer comments are as following:
Responses to the reviewer comments:
Comments and Suggestions for Authors
Authors generated and tested attenuated nGPV strain of ducklings.
The major minus is that authors did not perform sequencing on viruses especially parental strain and p35 mildly attenuated strain. From results it seems that still the p35 induces pathogenicity in ducklings. We do not know how many mutations and where in the genome were introduced, also in discussion it should be discuss possible reversion to WT. It is standard procedure to sequence genome of potential vaccine.
No proper statistical analysis was performed, you cannot see it on graphs and figures.
Therefore I recommend major revision.
In this manuscript, we attenuated the selected NGPV strains and characterized their pathogenicity in animals. In subsequent studies, we will sequence these strains and utilize the attenuated variants to evaluate their immune protective efficacy in animal models. Additionally, we are so sorry that we didn’t perform statistical analysis. All the data in this revised manuscript have been subjected to rigorous statistical significance testing.
General comment:
N-GPV i think NGPV or nGPV is more commonly used so no dash.
We referred to this reference “Chen Hao, Dou Yanguo, Tang Yi et al. Isolation and Genomic Characterization of a Duck-Origin GPV-Related Parvovirus from Cherry Valley Ducklings in China[J]. PLoS One, 2015, 10: e0140284.”, which reported this disease for the first time, and used N-GPV. So, we also used N-GPV in this manuscript.
Specific comment:
Q1: Line 31 up-regulate cange to increased (up regulat usually implies genes)
Revised. Line 32.
Q2: line 34 omit negative sense it is relevant for RNA viruses only
Revised. Line 34.
Q3: Figure 2 description in legend mixed up
Revised. Line137-139, Line146-151.
Q4: Figure 3 what is an arrow? Line 147-148 From the pictures I can not see any haemorhages. Fig 3 Apart from liver pictures, everything looks the same what is the point of this figure?
It represents hemorrhage in lung, liver, and spleen. Additionally, this disease leads to of stunted growth in ducks, so organs in experimental group look smaller than the control in figure 3. This figure looks pixelated due to compression in the PDF. Please loot at “author-coverletter-45603196.v1.docx”
Q5: Figure 5. Are this differences statistically important?
Yes. We are so sorry that we didn’t perfume statistical analysis. All the data in this revised manuscript have been subjected to rigorous statistical significance testing.
Q6: 5f wrong description in graph.
Revised. Line178-179.
Q7: Line 177what does it mean “red staining”
We have changed “red staining” to “hemorrhage”. Line 184.
Q8: Figure 6 what are the arrows? Also see general comment, arrowheads should be black, because it fades with background picture
These arrows represent histopathological changes in liver, spleen, duodenal and lung (Line 182-190). The color of arrows has been corrected.
Q9: Fig 7 again statistics! Add days p.i to graphs for quicker understanding for the reader
d- viremia (no blood shering)
All the data in the revised manuscript have been subjected to rigorous statistical significance testing. And “dpi” has been added to Figure 7a, 7b, 7c.
The results of N-GPV copy numbers in blood are shown in Figure. 7d. The virus copy number in the blood of NMG21-15 and NMG21-25 groups could be detected as early as 1 dpi and reached the replication peak at 14 dpi, while the other 3 groups reached the replication peak at 7 dpi. Line 217-220.
Q10: 238 this sentence is not clear :as a newly n_GPV?)
The sentence was deleted. Line 239.
Q11: In discussion you mention feather abnormalities, it should be mention in results section.
Feather loss was only observed in the NMG21-35 group (Figure. 2d) (Lines 138-139).
Q12: Discussion part does not discuss the results thorouly, what is the basic of attenuation (why there is antibody drop in day 14 in wt and other mutants).
The attenuation of a NMG21-35 strain might be the result of a combination of multiple mutations along the genome, and can occur via multiple molecular mechanisms. Further genetic investigation will be necessary to determine the key mutations responsible for the attenuation in NMG21-35 strain. Moreover, whether individual or a combination of genetic changes of N-GPV strains alter viral infectivity, pathogenicity and replication efficiency need further examination by reverse genetics technology. Line 271-276.
It was speculated that NMG21, NMG21-5 and NMG21-15 strains were more virulent, and secondary infection occurred at 14 dpi. The weaker virulence of the reinfected strain resulted in increased antibody levels in the ducklings.
Q13: Also some figures are unreadable, like anatomopathological lesions, histopathology, they are too small. Maybe do not show pictures from every challenge group, just write about it in manuscript and show magnifications.
This figure looks pixelated due to compression in the PDF. Pleases look at“author-coverletter-45603196.v1.docx”.

Reviewer 2 Report
Comments and Suggestions for Authors
Dear Authors,
The manuscript certainly touches on an important topic. The attenuated strain NMG21-35 demonstrated the ability to induce a stronger humoral antibody response upon inoculation. This finding suggests that this strain may be a valuable candidate for inclusion in vaccination programs, potentially mitigating the economic losses associated with N-GPV infections in the commercial duck farming industry. However, the manuscript cannot be published in its current form. It is necessary to smoothly lead the reader to the research problem. Specify how many viruses are known in the object of study. In the methods, the sample of experimental animals must be given clearly and justified. Many methodological aspects of organ damage must be described. Changes in the size of organs in the control and experiment also deserve detailed attention. In the photographs of the authors, the skull and beak of ducklings in the control and experiment are significantly different. I believe that the size of the skull should be indicated. The text of the manuscript in different chapters is mixed up and should be moved to the appropriate chapters. In many methodological aspects, the authors omit important information. It should be added. You should focus on writing the conclusion of the manuscript based on the hypothesis, there is no hypothesis yet, and there is no clear goal. Please correct all these shortcomings. The authors have obtained interesting results, presented them and they should be disclosed in the conclusions of the manuscript. After eliminating all the comments, the manuscript can be reviewed again.

Author Response
Dear Editor and Reviewers:
Many thanks for your professional evaluation of our manuscript entitled “Attenuation of an original novel goose parvovirus strain NMG21 via serial cell passage” (ID: ID: viruses-3573035). On behalf of my co-authors, we would like to express our great appreciation to the editor and reviewers.We studied the comments carefully and have made corrections that we hope meet with approval. The main corrections in the paper and the responses to the reviewer comments are as following:
Responses to the reviewer comments:
- There is no need for this in the title of the manuscript.
Revised. Line 2.
- Please report how many viruses are known in geese and then move on to the specific problem of study.
This disease leads to BADS in ducklings. Because of increasing amounts of ducks and geese, viral diseases showed an increasing trend. There are too many duck and goose viral diseases (avian influenza virus, duck tembusu virus, newcastle disease virus) to list. This study performed epidemiological investigation of infectious diseases in geese, “He Dalin, Wang Fangfang, Zhao Liming et al. Epidemiological investigation of infectious diseases in geese on mainland China during 2018-2021[J]. Transbound Emerg Dis, 2022, 69: 3419-3432.”
- clarification required
Revised. Line 19.
- please remove one of the words so there are no repetitions
Revised. Line 20.
- please remove one of the words so there are no repetitions
Revised. Line 20.
- It is better to replace the keyword with another word to cover as wide a range of interests as possible.
Revised. We replaced “attenuated variant” with “virulence”.
- The fight against zoonotic viral infections of wild and domestic animals occupies an important place in human activities [Germeraad at el. 2019, Andreychev et al. 2022].
Revised. Line27-28.
- You need to describe in detail what you mean.
The sentence was deleted. Line 33.
- Please specify
Revised. Line 40-41.
- This is somewhat contradictory to the first sentence of this paragraph. Please rearrange the sentences so that there is a logical connection.
Th first sentence has been deleted. GPV and N-GPV are two different waterfowl disease viruses. Given that the GPV vaccine and anti-GPV yolk antibodies have some efficacy in protecting ducks from BADS, this may be the main reason for the current lack of an effective vaccine to prevent and control BADS.
- This idea was already mentioned earlier in the text. There is no need for it here.
Revised. Line 45.
- This needs to be changed. Please add a hypothesis and clearly formulate the aim of the study.
Revised. Line 46-47.
- In what quantity?
Revised. Line 55-56.
- How is it with this disease? Why from there?
This NMG21 strain was preserved in our laboratory, which shows better pathogenicity and virulence to ducks and duck embryos. What’s more, the virus titer keeps a higher level. We conducted this animal experiment in Shandong Agricultural University, which is located in Tai’an, Shandong Province, so we purchased ducks and embryos in Tai’an. Line 60-62.
- Please add these conditions in detail.
Based on this reference “Chen Hao, Dou Yanguo, Tang Yi et al. Isolation and Genomic Characterization of a Duck-Origin GPV-Related Parvovirus from Cherry Valley Ducklings in China[J]. PloS One, 2015, 10: e0140284”, we conducted this animal experiment. All the breeding conditions are the same as this reference. Line 60.
- Is this the first methodological part you have conducted or are there publications? If there are publications, then it is necessary to provide references to the sources of literature.
No. This is the first methodological part we have conducted.
- How? Add an explanation
It means atrophy of duck embryo bodies. Line 75.
- This should be stated earlier in the text.
Based on your suggestions, we stated this in 2.1.
- Please specify
Revised. Line 84-85.
- Specify the volumes of each organ required for analysis.
Volume of tissue specimens is 1cm×1cm×1cm. Line 94.
- Please move this to Materials and Methods
Revised. Line 79-80.
- In this photo (f) it is clear that the beak and skull of the control goose are larger than those of the experimental specimens. Therefore, the size of the skull, as well as the shortness of the beak, may be important in the analysis? At least this visual information should be explained in the text.
Revised. Line 137-141, line 146-151.
- How much? At what rate did pathogenicity decrease over time?
The details and results are shown in Table 1.
Table 1 The virus titers and virulence of different serial passages of NMG21 strain
|
|
P0 (NMG21) |
P5 (NMG21) |
P15 (NMG21) |
P25 (NMG21) |
P35 (NMG21) |
|
EID50(0.2mL) |
10-7 |
10-5 |
10-4.7 |
10-4.2 |
10-3.7 |
|
Embryonic mortality rate |
33% |
30% |
8% |
3% |
0% |
|
Embryonic change |
development retardation and beak atrophy |
severe hemorrhaging and beak atrophy |
free of visible lesions and beak atrophy |
free of visible lesions and normal beak |
free of visible lesions and normal beak |
|
Duckling mortality rate |
47% |
40% |
26.7% |
6.7% |
1.2% |
|
Duckling symptom |
paralysis, short beak, tongue swelling, and growth retardation, |
paralysis, short beak, tongue swelling, and growth retardation, |
paralysis, short beak, tongue swelling, and growth retardation, |
growth retardation, |
Feather loss |
|
Liver hemorrhage rate |
93% |
73% |
37.3% |
10% |
0% |
|
Pulmonary hemorrhage rate |
93% |
80% |
26.7% |
13% |
0% |
|
Splenic hemorrhage and swelling rate |
100% |
80% |
43.3% |
16.7% |
0% |
- Indicate the variation of each trait in the control group.
The line was derived from the control group data, while the colored points correspond to the experimental group. Figure 4, line 170.
- Please highlight the organs that are most affected.
Revised. Line 182-183.
- Where appears earlier indicate
Revised. Line 215, line 217-220.
- Is this necessary?
Revised. This sentence is deleted. Line 223.
- Are there similar studies on them? If so, it is necessary to compare the results in detail.
No. They are different diseases.
- There is a need to move this to the Introduction
Revised. Line 38-40.
- You already wrote this in the manuscript. No need to repeat it.
Revised. This sentence is deleted.
- You already wrote this in the manuscript. No need to repeat it.
Revised. This sentence is deleted.
- No need
Revised. This sentence is deleted.
- No need
Revised. This sentence is deleted.
- This can be used when formulating a hypothesis.
These sentences have been deleted.
- Here you need to indicate your practical recommendations and subsequent consistent actions for applying the results for veterinary medicine.
There is no effective treatment method for this disease at present, it’s quite important to prevent this disease. Although at present, some duck farmers use GPV egg yolk antibody to treat this disease, there is no commercialized biological product for N-GPV infection in ducks with a lot number. In subsequent studies, we will utilize the attenuated variants to evaluate their immune protective efficacy in animal models.
- Add: Germeraad, E. A.; Sanders, P.; Hagenaars, T. J.; de Jong, M. C.; Beerens, N.; Gonzales, J. L. Virus shedding of avian influenza in poultry: a systematic review and meta-analysis. Viruses 2019, 11, 812, doi:10.3390/v11090812.
Andreychev, A.; Boyarova, E.; Brandler, O.; Tukhbatullin, A.; Kapustina, S. Terrestrial and subterranean mammals as reservoirs of zoonotic diseases in the central part of European Russia. Diversity 2022, 15, doi:10.3390/D15010039.
We have added these two references to this manuscript. Reference 1, 2.

Round 2
Reviewer 1 Report
Comments and Suggestions for Authors
none
Reviewer 2 Report
Comments and Suggestions for Authors
Dear Authors,
I am satisfied with the revision of the manuscript. You have supplemented the manuscript with data and made corrections. The title of the manuscript now better reflects the content of the article. Explanations have been added in the text on the attenuation of the new strain of goose parvovirus NMG21 by means of a sequential 2-cell passage. The results of the experimental and control samples are properly substantiated. The review of the research results was chosen appropriately, as were the statistical methods used for its analysis. The article takes into account the comments on the methodology. The analysis and conclusion for each chapter are sufficient and do not raise objections. References to sources of literature have been adjusted. The results of previous studies by other authors have been taken into account. I recommend it for the journal Viruses.